# SMILe : Scalable Meta Inverse Reinforcement Learning through Context-Conditional Policies

**Seyed Kamyar Seyed Ghasemipour**
University of Toronto
Vector Institute
kamyar@cs.toronto.edu

**Shixiang Gu**
Google Brain
shanegu@google.com

**Richard Zemel**
University of Toronto
Vector Institute
zemel@cs.toronto.edu

## Abstract

Imitation Learning (IL) has been successfully applied to complex sequential decision-making problems where standard Reinforcement Learning (RL) algorithms fail. A number of recent methods extend IL to few-shot learning scenarios, where a meta-trained policy learns to quickly master new tasks using limited demonstrations. However, although Inverse Reinforcement Learning (IRL) often outperforms Behavioral Cloning (BC) in terms of imitation quality, most of these approaches build on BC due to its simple optimization objective. In this work, we propose SMILe, a scalable framework for Meta Inverse Reinforcement Learning (Meta-IRL) based on maximum entropy IRL, which can learn high-quality policies from few demonstrations. We examine the efficacy of our method on a variety of high-dimensional simulated continuous control tasks and observe that SMILe significantly outperforms Meta-BC. Furthermore, we observe that SMILe performs comparably or outperforms Meta-DAgger, while being applicable in the state-only setting and not requiring online experts. To our knowledge, our approach is the first efficient method for Meta-IRL that scales to the function approximator setting. For datasets and reproducing results please refer to https://github.com/KamyarGh/rl_swiss/blob/master/reproducing/smile_paper.md.

## 1 Introduction

Modern advances in reinforcement learning aim to alleviate the need for hand-engineered decision-making and control algorithms by designing general purpose methods that learn to optimize provided reward functions. In many cases however, it is either too challenging to optimize a given reward (e.g. due to sparsity of signal), or it is simply impossible to design a reward function that captures the intricate details of desired outcomes. One approach to overcoming such hurdles is Imitation Learning (IL) (also known as Learning from Demonstrations (LfD)) [29, 28, 21], where algorithms are provided with expert demonstrations of how to accomplish desired tasks.

In practical applications such as robotics we are interested in learning policies for accomplishing a variety of tasks with very few demonstrations. As a result, a number of recent works study IL in the context of few-shot or meta learning [3, 7, 35, 33]. By leveraging the shared structure among tasks, these approaches can learn a generic policy that adapts quickly to related new tasks using small sets of new demonstrations.

Despite the importance and popularity of Meta-IL, existing approaches [3, 7, 35] mostly rely on Behavioral Cloning (BC) [28], where learning from demonstrations is treated as a supervised learning problem and policies are trained to regress expert actions from a dataset of expert demonstrations. On the other hand, much less work explores meta learning through the lens of Inverse Reinforcement Learning (IRL) [21, 1], whose aim is to infer the reward function of the expert, and subsequently train a policy to optimize this reward. In single-task benchmarks for continuous control [2], recent IRL

methods [17, 9] have been shown to outperform BC by a wide margin, particularly in the low data regime where a very limited number of expert trajectories are available. Furthermore, IRL algorithms can be employed even in settings where we only observe the expert's states, and not their actions. As a result, within the problem of Meta-IL, it is desirable to develop scalable methods for Meta-IRL in particular. However, due to IRL algorithms being significantly more complex than their BC counterparts, prior approaches to Meta-IRL have either failed to scale to the function approximator setting [34], or require pre-training procedures that are beyond state-of-the-art and easily become computational bottlenecks [33].

In this work, we interpret the problem of Meta-IRL as IRL conditioned upon a context, where the context takes the form of expert demonstrations. We present an efficient method that scales to intractable function approximators, and demonstrate its performances on difficult high-dimensional continuous control environments. We call our algorithm Scalable Meta-Inverse reinforcement Learning, or SMILe. Reflecting the results from prior work in the single-task setting, we observe that even with exceedingly large amount of demonstrations Meta Behavioral Cloning performs very poorly. However, even in the state-only setting - where expert actions are unobserved - and with a very limited set of demonstrations, our proposed Meta-IRL approach can achieve close to expert performance. Additionally, as our method is not dependent on any particular form of context, our work opens avenues for Meta-IRL conditioned on different forms of data such as linguistic task descriptions or images of desired goal states.

## 2   Background

In this section we discuss relevant background on Max-Ent IRL and modern approaches to this problem, which our approach to meta IRL builds from.

### 2.1   Maximum Entropy Inverse Reinforcement Learning

Consider a Markov Decision Process (MDP) represented as a tuple $(\mathcal{S}, \mathcal{A}, \mathcal{P}, r)$ with state-space $\mathcal{S}$, action-space $\mathcal{A}$, dynamics $\mathcal{P}$, reward function $r(s, a)$. In Maximum Entropy (Max-Ent) reinforcement learning [30, 32, 26, 8, 15, 16], the goal is to find a policy $\pi$ such that trajectories sampled using this policy follow the distribution,

$$p(\tau) = \frac{1}{Z}\exp(R(\tau))p(s_0)\prod_t p(s_{t+1}|s_t, a_t)$$

where $\tau = (s_0, a_0, s_1, a_1, ...)$ denotes a trajectory, $R(\tau) = \sum_t r(s_t, a_t)$, and $Z = \int_\tau \exp(R(\tau))$. Hence, trajectories that accumulate more reward are exponentially more likely to be sampled, provided they have similar likelihood under the MDP dynamics.

Converse to the standard RL setting, in Max-Ent IRL [37, 36] we are instead presented with an optimal policy $\pi^{\exp}$ - or more realistically sample trajectories from such a policy - and we seek to find a reward function $r$ that maximizes the likelihood of the trajectories sampled from $\pi^{\exp}$. Formally, the simplified form of the objective is: $\max_r \mathbb{E}_{\tau \sim \pi^{\exp}}[R(\tau) - \log Z]$. As in all energy-based models, optimizing a maximum likelihood learning objective is difficult due to the need to estimate the partition function. Initial methods addressed this problem using dynamic programming [37, 36], and recent approaches present methods aimed at intractable domains with unknown dynamics [6, 17, 5, 9, 20].

### 2.2   Adversarial Max-Ent IRL

Instead of recovering the expert's reward function and subsequently searching for the optimal policy, recent successful methods in Max-Ent IRL aim to directly recover the policy that would result from this process. Since such methods only recover the policy, it would be more accurate to refer to them as Imitation Learning algorithms.

**AIRL**   [17] present one of the first formulations of Max-Ent IRL, with a successful special case dubbed GAIL. Subsequent to the advent of GAIL [17], [5] present a theoretical discussion of an adversarial training method for Max-Ent IRL that could recover the Max-Ent reward function from

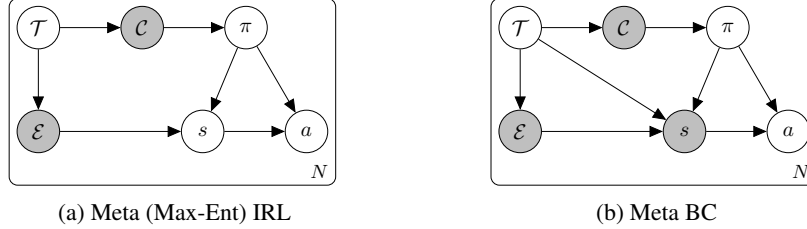

(a) Meta (Max-Ent) IRL          (b) Meta BC

Figure 1: Graphical models describing the generative process in Meta-IRL and Meta-BC. $\mathcal{E}$ represents the initial state distribution and dynamics (i.e. the environment) for task $\mathcal{T}$, and $\mathcal{C}$ represents a context. Grey and white nodes denote observed and unobserved variables respectively. In Meta-IRL the aim is to match the state-action marginal of a context-conditioned policy to that of the task expert, whereas in Meta-BC the aim is to match expert actions in states visited by the expert.

expert demonstrations and simultaneously train the Max-Ent policy corresponding to that reward. [9] present a practical implementation of this method, dubbed Adversarial Inverse Reinforcement Learning (AIRL), alongside additional contributions that are not directly relevant to our work. The method for AIRL is formulated as follows:

Let $\rho^{\text{exp}}(s, a), \rho^{\pi}(s, a)$ denote the state-action marginal distributions of the expert and student policy respectively and let $D(s, a) : \mathcal{S} \times \mathcal{A} \to [0, 1]$ be the discriminator. AIRL defines the reward function, $r(s, a) := \log D(s, a) - \log (1 - D(s, a))$, and sets the objective for the student policy to be the RL objective, $\max_{\pi} \mathbb{E}_{\rho^{\pi}(s,a)} \left[ \sum_t r(s_t, a_t) \right]$. This leads to an iterative optimization process alternating between optimizing the discriminator and the policy, and it is shown that this procedure implicitly optimizes the Max-Ent IRL objective.

**Performance With Respect to BC**     Methods such as GAIL and AIRL have demonstrated significant performance gains compared to BC. In particular, in standard Mujoco benchmarks [31, 2], direct methods for Max-Ent IRL achieve strong performance using limited amount of demonstrations from an expert policy, an important failure scenario for BC.

## 2.3    Max-Ent IRL as Distribution Matching

As noted above, [17] present a class of methods for Max-Ent IRL that directly retrieve the expert policy without explicitly finding the reward function of the expert. They demonstrate that the Max-Ent IRL problem is equivalent to matching $\rho^{\pi}(s, a)$ to $\rho^{\text{exp}}(s, a)$, and that GAIL minimizes the Jensen-Shannon divergence between these two distributions. In concurrently published work, we have demonstrated that the AIRL [9] objective optimizes the reverse KL divergence $\text{KL}\left(\rho^{\pi}(s, a) || \rho^{\text{exp}}(s, a)\right)$. Using a connection to $f$-GANs [23], we presented a generalization of AIRL for minimizing *any* $f$-divergence between the two distributions.

The objective for standard Behavioral Cloning, $\max_{\pi} \mathbb{E}_{\rho^{\text{exp}}(s,a)} \left[ \log \pi(a|s) \right]$, does not depend on the states visited by the student policy and only encourages the policy's actions to match those of the expert in the states visited *by the expert*. At test-time, due to not exactly matching the expert, BC policies will visit new states for which they did not receive expert supervision. This covariate shift results in compounding errors in subsequent timesteps and has been identified as one of the most important factors contributing to Behavioral Cloning's poor performance [27]. In contrast, the results from prior work discussed above demonstrate that the core defining characteristic of Max-Ent IRL is not its interpretation of recovering the expert's reward function, but instead the matching of $\rho^{\text{exp}}(s, a)$ and $\rho^{\pi}(s, a)$. This encourages the imitation policy to learn to visit similar states as the expert.

## 3    Meta Inverse Reinforcement Learning

### 3.1    Problem Formulation

We begin by framing the Meta-IRL problem we seek to solve. Let $p(\mathcal{T})$ denote a distribution of tasks we are interested in acquiring a policy for. Per task, we have its initial state and transition distributions (i.e. the environment for that task), which we denote using the variable $\mathcal{E}$. $p(\mathcal{C}|\mathcal{T})$ represents a

distribution over contexts for task $\mathcal{T}$. Contexts for a task can take many forms; they can be a set of demonstrations, language instructions, or even an image representing a desired goal state. Given access to an environment $\mathcal{E}$, and provided with a context $\mathcal{C}$, a context-conditional policy induces a joint distribution over states and actions, $\rho^\pi(s, a | \mathcal{C}, \mathcal{E})$. This generative process can be described using the graphical model shown in figure 1a. Let $\rho^{\exp}(s, a | \mathcal{C}, \mathcal{E}) = \int_\mathcal{T} p(\mathcal{T} | \mathcal{C}, \mathcal{E}) \rho^{\exp}(s, a | \mathcal{T}, \mathcal{E})$ denote the posterior expert state-action distribution. Building on the discussion in section 2.3 our Meta-IRL objective takes the form,

$$\min_\pi \mathbb{E}_{p(\mathcal{C},\mathcal{E})} \Big[ M \Big( \rho^\pi(s, a | \mathcal{C}, \mathcal{E}) \quad || \quad \rho^{\exp}(s, a | \mathcal{C}, \mathcal{E}) \Big) \Big] \tag{1}$$

where $M$ is any measure of discrepancy between distributions[1]. Intuitively, this objective requires the policy to match expert behaviour for different tasks, proportional to the likelihood of each task given the context and environment. The counterpart Meta-BC objective takes the following form,

$$\min_\pi \mathbb{E}_{p(\mathcal{C},\mathcal{E})} \Big[ \mathbb{E}_{\rho^{\exp}(s|\mathcal{C},\mathcal{E})} \Big[ M \Big( \rho^\pi(a | s, \mathcal{C}, \mathcal{E}) \quad || \quad \rho^{\exp}(a | s, \mathcal{C}, \mathcal{E}) \Big) \Big] \Big] \tag{2}$$

## 3.2 Approach

Inspired by AIRL [9], the specific form of objective 1 we choose to optimize is,

$$\min_\pi \mathbb{E}_{p(\mathcal{C},\mathcal{E})} \Big[ \mathrm{KL} \Big( \rho^\pi(s, a | \mathcal{C}, \mathcal{E}) \quad || \quad \rho^{\exp}(s, a | \mathcal{C}, \mathcal{E}) \Big) \Big] \tag{3}$$

Similar to AIRL, we employ the following iterative optimization scheme,

$$\max_D \mathbb{E}_{p(\mathcal{C},\mathcal{E})} \Big[ \mathbb{E}_{\rho^{\exp}(s,a|\mathcal{C},\mathcal{E})} [\log D(s, a, \mathcal{C}, \mathcal{E})] + \mathbb{E}_{\rho^\pi(s,a|\mathcal{C},\mathcal{E})} [\log (1 - D(s, a, \mathcal{C}, \mathcal{E}))] \Big] \tag{4}$$

$$\max_\pi \mathbb{E}_{p(\mathcal{C},\mathcal{E})} \Big[ \mathbb{E}_{\tau \sim \pi(\cdot|\mathcal{C},\mathcal{E})} \Big[ \sum_t \log D(s_t, a_t, \mathcal{C}, \mathcal{E}) - \log (1 - D(s_t, a_t, \mathcal{C}, \mathcal{E})) \Big] \Big] \tag{5}$$

where the discriminator model is conditioned upon the context and environment, and $\tau \sim \pi(\cdot | \mathcal{C}, \mathcal{E})$ denotes sampling trajectories in environment $\mathcal{E}$ using the policy conditioned on context $\mathcal{C}$. As in the AIRL formulation, the objective for each conditional policy is an RL objective using a reward defined by the discriminator. Many practical considerations are necessary in order to convert equations 4 and 5 into an efficient and effective algorithm. The following sections outline such practicalities.

## 3.3 Optimizing the Discriminator Objective

**Conditioning on $\mathcal{E}$**    In addition to conditioning on the state, action, and context, the discriminator must take into account the environment $\mathcal{E}$. However, as there does not exist a convenient representation for an environment, we decide to not condition the discriminator model on this variable. This choice can be particularly hindering in settings where the different tasks are defined by their varying dynamics. In such scenarios we rely on the context variable $\mathcal{C}$ conveying the necessary information regarding the environment. For example, when the context consists of a set of expert demonstrations, it is reasonable to assume we can acquire information about the dynamics through observing the expert's trajectories.

**Monte-Carlo Estimation**    We would like to optimize the discriminator objective using Monte-Carlo estimates. To do so, recalling that $\rho^{\exp}(s, a | \mathcal{C}, \mathcal{E}) = \int_\mathcal{T} p(\mathcal{T} | \mathcal{C}, \mathcal{E}) \rho^{\exp}(s, a | \mathcal{T}, \mathcal{E})$, and removing $\mathcal{E}$ from the discriminator's input, we can rewrite equation 4 as,

$$\mathbb{E}_{p(\mathcal{T},\mathcal{C},\mathcal{E})} \Big[ \mathbb{E}_{\rho^{\exp}(s,a|\mathcal{T},\mathcal{E})} [\log D(s, a, \mathcal{C})] - \mathbb{E}_{\rho^\pi(s,a|\mathcal{C},\mathcal{E})} [\log (1 - D(s, a, \mathcal{C}))] \Big] \tag{6}$$

In this form we can see how to obtain estimates of the discriminator objective. After obtaining a sample task, context, and environment, the first term can be estimated by sampling state-action pairs through rollouts of the task's expert in the given environment. Similarly, the second term can be estimated through rollouts of the conditional policy.

**Off-Policy Training**    An important computational bottleneck in the GAIL [17] and AIRL [9] algorithms is that the discriminator and policy must be trained with on-policy samples. This is a large computational burden since after every update to the policy we must generate multiple trajectories in order to train the discriminator. The meta-learning setting exacerbates this problem as we must now generate multiple trajectories for many tasks. In this work we take inspiration from [20] and instead maintain a replay buffer of policy rollouts per task. We then estimate the second term of equation 6, by sampling a task $\mathcal{T} \sim p(\mathcal{T})$, sampling $\mathcal{C}, \mathcal{E} \sim p(\mathcal{C}, \mathcal{E}|\mathcal{T})$, and finally sampling state-action pairs from the replay buffer of task $\mathcal{T}$. The main effect of this approach is that the discriminator now estimates the likelihood ratio between the experts and the *mixture of policies that populated the replay buffers*.

## 3.4    Optimizing the Policy Objective

**Obtaining a Context Representation**    In order to optimize their respective objectives, both the policy and the discriminator models must learn to use provided contexts. The naïve approach would be to allow the discriminator and policy to learn their own representations. However, since the discriminator is trained using state-action pairs from *both* the experts as well as the policy, it is much better poised for learning effective representations of the context. On the other hand, the policy must learn to extract important aspects of the context solely through the reward signal it obtains from the discriminator. As a result, we choose to allow the discriminator to train its own context-encoder model, and have the policy adapt itself to the discriminator's representation of the context. Not only does this design choice remove the computational cost of training a second encoder, it also leads to a stable and effective learning algorithm.

**Off-Policy Training**    As mentioned above, we would like to leverage off-policy training for improved sample-efficiency. To correctly optimize the policy objective using samples from the replay buffer, importance sampling must be employed. However, it can be quite challenging to accurately estimate the necessary importance weights. [20] demonstrate that omitting these weights works well in practice. We follow the same approach and obtain the sample-efficiency of off-policy training without empirical hindrance to policy performance.

## 3.5    Algorithm

Taking considerations above into account, we arrive at the final form of our approach, SMILe . In this section we present a conceptual overview of SMILe and defer exact details to Algorithm 1 in Appendix A. The SMILe training procedure alternates between generating rollouts and updating models.

**Generating Rollouts**    In each iteration we add $m$ trajectories to the task replay buffers by sampling $m$ tasks from the meta-train set, sampling a context for each task, and generating a trajectory using the context-conditioned policies in the corresponding environments. After generating the rollouts, for $d$ times we alternate between updating the discriminator and the policy.

**Updating the Discriminator**    To update the discriminator model, including its context encoder, we first obtain a minibatch of meta-train tasks as well as a context for each. For a given task $\mathcal{T}$, positive samples for the discriminator objective have the form $(s, a, \mathcal{C}_{\mathcal{T}})$, where $(s, a)$ is a state-action pair sampled from $\mathcal{T}$'s expert demonstrations, and $\mathcal{C}_{\mathcal{T}}$ is the task context. Similarly negative batches are formed by concatenating $(s, a)$ pairs from each task's replay buffer with corresponding contexts.

**Updating the Policy**    Throughout this work we will perform policy updates using the Soft-Actor-Critic (SAC) algorithm [16]. In this step, we first sample a minibatch of tasks and contexts, and obtain a representation for each context using the discriminator's encoder. To form an update batch for the policy, Q, and value functions, we sample $(s, a, s')$ transitions from each task's replay buffer and concatenate the context representation to the states. Subsequently, an SAC update step is done by using the rewards defined through the discriminator. We do not update the encoder in this step.

An important advantage of IRL compared to BC is that it also enables Imitation Learning using state-only demonstrations. In such settings the discriminator model takes as input state-next-state pairs $(s, s')$ instead of state-action pairs $(s, a)$.

| Method | 4 demos, 20 sub | 16 demos, 20 sub | 64 demos, 20 sub | 64 demos, 1 sub |
|---|---|---|---|---|
| Meta-BC | $0.931 \pm 0.620$ | $0.643 \pm 0.410$ | $0.468 \pm 0.274$ | $0.340 \pm 0.188$ |
| SMILe (State-Action) | $\mathbf{0.181 \pm 0.095}$ | $0.125 \pm 0.060$ | $\mathbf{0.102 \pm 0.100}$ | $0.118 \pm 0.078$ |
| SMILe (State-Only) | $0.186 \pm 0.107$ | $\mathbf{0.113 \pm 0.058}$ | $0.121 \pm 0.086$ | $0.138 \pm 0.074$ |
| Meta-DAgger | — | — | — | $\mathbf{0.082 \pm 0.059}$ |

Table 1: Mean and standard deviation of absolute difference between target velocity and velocity of trained models (lower is better). "$n$ demos, $m$ sub" indicates that $n$ expert trajectories were provided *per task*, with each trajectory uniformly subsampled by a factor of $m$ starting at a random offset. As a reminder, meta-test target velocities range from $[0.05, 2.95]$. Even with a very large number of demonstrations per task, Meta-BC obtains velocities quite distant from the target, whereas with quite few demonstrations, both variants of Meta-IRL obtain respectable performance. Meta-DAgger outperforms SMILe in this experiment, at the expense of much larger number of queries from an online expert. Full evaluation details in appendix C.

## 4 Experiments

As discussed above, contexts may take many forms. In this work, however, we focus exclusively on the Meta Imitation Learning problem and use expert trajectories as contexts. Through our experiments we seek to empirically evaluate the following questions:

- Can SMILe significantly outperform Meta-BC, analogous to the single-task setting?
- Can expert demonstrations be an effective source of information about dynamics?
- Can SMILe be an effective learning algorithm when it is necessary to combine information from multiple demonstration trajectories?

To answer these questions we compare state-action and state-only variants of SMILe to a strong Meta-BC baseline on a varied range of simulated continuous control problems. For each experiment the Meta-BC baseline model is obtained by composing SMILe's context encoder and policy models. We also compare SMILe to a meta-learning variant of DAgger [27], a Behavioural Cloning algorithm designed to overcome the covariate shift problems associated with standard BC. Meta-DAgger requires online access to the experts, and hence is not directly comparable to SMILe. Nonetheless, it can serve as an informative on the upper bound on the performance we may expect from standard BC. Full baseline model and training details are presented in Appendix B. In all experiments below we use the same form of context encoder model inspired by permutation-invariant encoders used in recent meta-learning literature [10, 13]. This modelling choice is discussed in Appendix A.

### 4.1 HalfCheetah Random Velocity

The HalfCheetah Random Velocity task is a popular baseline for meta-learning in standard RL. To evaluate SMILe, we adapt this task for the Few-Shot Imitation Learning setup. Each task is defined by a target velocity that we wish a HalfCheetah agent maintain over the duration of an episode; episodes are of length 1000 and start with the agent at standstill. Target velocities for meta-train tasks range from 0 to 3, uniformly spaced at 0.1 intervals, and meta-test tasks are defined by the range 0.05 to 2.95, uniformly spaced at 0.1 intervals. In this experiment, a sampled context is a random number (1 to 4) of expert trajectories. To obtain expert demonstrations, we train an expert policy using Soft-Actor-Critic [16] which observes as part of the state the desired target velocity. We train all models using various amounts of total expert demonstrations and evaluate on the meta-test tasks using context trajectories generated by the pre-trained expert. Table 1 presents evaluation results. As can be seen both variants of Meta-IRL *very significantly* outperform Meta-BC: With the least amount of demonstrations, Meta-IRL is able to achieve respectable performance whereas Meta-BC obtains quite poor results even with the exceedingly large maximum amount of demonstrations used. In this experiment we observe that Meta-DAgger can outperform SMILe, at the expense of much larger number of queries from an online expert.

### 4.2 Ant 2D Goal

The second set of tasks we experiment with is 2D navigation using a simulated Ant agent [2]. The agent starts each episode in the center of the environment at coordinates (0,0) and has to navigate to a certain point on the perimeter of a circle with radius 2.0 centered at the origin. Episodes have

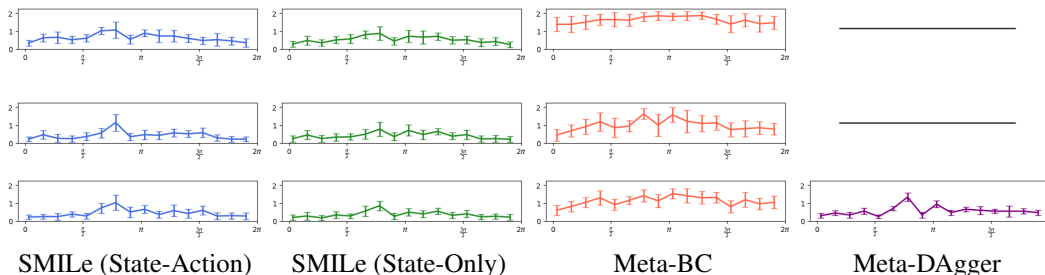

| SMILe (State-Action) | SMILe (State-Only) | Meta-BC | Meta-DAgger |

Figure 2: Mean and standard deviation of minimum distance when attempting to reach meta-test targets (lower is better) on the Ant 2D tasks. x-axis denotes angle of the target on the circle, and y-axis denotes how closely the agent approached the target. Rows from top to bottom in order present results when trained on 4, 16, and 64 demonstrations *per task*. Despite its stronger assumptions, Meta-DAgger performs comparably to SMILe. Full evalution details in Appendix D.2. In Appendix D.3 we also demonstrate that RL is not sufficient for learning this task and that Imitation Learning is necessary to overcome the exploration problem.

a horizon of 100 timesteps. The meta-training set consists of 32 target positions located at every integer multiple of $\frac{\pi}{16}$ radians on the circle. The meta-testing set consists of 16 targets located at every $\frac{2n\pi}{16} + \frac{\pi}{32}$ angle on the circle. In this experiment contexts are single expert trajectories. To obtain expert demonstrations, we found it challenging to train a single expert policy that was uniformly good at navigating to the different targets. As a result, we trained separate experts for each train and test target location.

Figure 2 presents results when training on 4, 16, and 64 demonstrations per meta-train task (4 random seeds per model per setting). As can be seen, both variants of our method significantly outperform Meta-BC: models trained with Meta-IRL learn to approach most meta-test target locations well, whereas Meta-BC models barely get close to most targets. We also observe that despite its stronger assumptions Meta-DAgger performs comparably to both variants of SMILe.

### 4.3 Handling Varying Dynamics

In section 3.3 we described that we do not condition directly on the environment variable $\mathcal{E}$ and rely on contexts when it is necessary to take dynamics into account. In this section we evaluate the feasibility of this approach by designing an experiment where tasks are defined by their varying dynamics. Specifically, we begin with the Walker environment [2] and generate different task by randomizing various dynamics variables[2]. We use 50 meta-train tasks and perform evaluations on 25 meta-test tasks. As in the previous section we train separate expert policies for each of the meta-train and meta-test tasks.[3] Per task we generate 32 expert trajectories (with a subsampling factor of 20) and the contexts used are single expert trajectories.

Despite having separate experts for each task, we continue to observe large variability in expert rewards obtained for each environment. This leads us to believe that some dynamics settings produce intrinsically harder tasks. As a result, we introduce the following custom evaluation metric. First, we trained a policy using RL on the unmodified Walker environment. Let $R_{\text{base}}^i$ denote its performance on task $i$. Additionally, let $R_{\text{expert}}^i$ denote expert performance for task $i$. For a trained meta-policy $\pi$, the evaluation metric we propose is: $\texttt{Eval}(\pi) := \frac{1}{N} \sum_{i=1}^{N} \frac{R_\pi^i - R_{\text{base}}^i}{R_{\text{expert}}^i - R_{\text{base}}^i}$ where $R_\pi^i$ is the performance of the meta-policy on task $i$, estimated by sampling multiple contexts for task $i$ and averaging the performance of the respective conditional policies.

Table 2 (Left) presents evaluations using our custom metric as well as Walker rewards.[4] Indeed we observe that policies trained with SMILe learn to adapt to new dynamics through conditioning on a single expert trajectory, whereas Meta-BC policies fail to outperform a baseline policy trained *only* on the unmodified environment.

| Method | Returns | Custom Metric | Success Rate | No-Op Rate |
|---|---|---|---|---|
| **Meta-BC** | $1289.9 \pm 63.0$ | $-0.403 \pm 0.054$ | $0.50 \pm 0.00$ | $\mathbf{0.00 \pm 0.00}$ |
| **SMILe (State-Action)** | $3106.6 \pm 67.6$ | $0.710 \pm 0.036$ | $\mathbf{0.68 \pm 0.01}$ | $0.02 \pm 0.02$ |
| **SMILe (State-Only)** | $\mathbf{3159.3 \pm 84.9}$ | $\mathbf{0.738 \pm 0.061}$ | $0.65 \pm 0.01$ | $0.01 \pm 0.00$ |
| **Meta-DAgger** | $2404.1 \pm 91.7$ | $0.275 \pm 0.089$ | $0.67 \pm 0.00$ | $0.14 \pm 0.01$ |

Table 2: (Left) Walker random dynamics. Results demonstrate that policies trained with SMILe learn to adapt to varying meta-test dynamics by conditioning on single expert trajectories. Note that the custom metric compares models to expert policies trained separately for each dynamics setting. (Right) Ant Linear Classification. Results show that while policies trained with Meta-BC learned the locomotion aspect of the task, they failed to connect their movements to the linear classification problem.

Interestingly, we observe that despite using orders of magnitude more queries from online task experts, Meta-DAgger cannot compete with SMILe. We believe this demonstrates the strong advantage of matching state-marginals using Inverse Reinforcement Learning (Section 2.3 and Figure 1): Due to the complexity of handling varying dynamics, not matching expert actions well can result in the Meta-DAgger policies quickly tumbling over. In contrast, to match the expert state distributions the SMILe policies will directly try to not tumble over, and can hence reach the later states.

## 4.4 Aggregating Information Across Demonstrations

In the HalfCheetah experiment described in section 4.1 contexts could be between 1 to 4 expert trajectories. Figure 4 indicates that for this task very little gain is made from conditioning on increasing number of demonstrations. As a result, in this section we present an experiment where combining information across multiple trajectories is *necessary* for good policy performance; we train a Mujoco Ant model to learn few-shot linear classification in 4 dimensions.

Each task is defined by a 4-dimensional hyperplane passing through the origin. For a given task, the Ant starts each episode at position $(0, 0)$ and observes as part of its state an ordered pair of 4-dimensional vectors sampled from $[-1, 1]^4$, one from each side of the hyperplane. To make a classification decision, the Ant must navigate to one of two target locations positioned at (1.41,1.41) and (-1.41, 1.41), depending on whether it believes the first vector is from the positive or negative class. Contexts are a random number of expert demonstrations (1 to 8), and a classification decision is made when the Ant is within 0.5 radius of either target. Each episode is 50 timesteps long. To generate expert demonstrations we scripted a policy to manually drive two of the experts trained for the "Ant 2D Goal" experiment. We trained using 64 tasks with 16 demonstrations for each, and evaluated models using 32 meta-test tasks.

Table 2 (Right) demonstrates that policies trained with SMILe can successfully combine information from multiple trajectories in order to perform classification, while Meta-BC performs at random. Interestingly, given the large number of demonstrations for only two target locations, the poor performance of Meta-BC models is not due to locomotion failure; it is due to their inability to connect locomotion to the classification task at hand. Lastly, we observe that Meta-DAgger achieves a similar success rate to SMILe, yet the number of episodes in which a classification decision is not made is significantly higher than all other methods.

## 5 Related Work

In this section we situate our work with respect to prior meta-learning algorithms for inverse RL, specifically those geared towards the Few-Shot setting and scaling to intractable environments.

[12] present two methods for Meta-IRL, one extending the maximum causal entropy formulation [36], and the other extending the Adversarial IRL (AIRL) formulation [9]. To adapt AIRL to the Few-Shot setting, the discriminator is optimized using the Reptile meta-learning method [22]. Every time a new task is sampled, the discriminator and policy are reinitialized and reoptimized with the hope being that the due to Reptile, the intialization for the discriminator will lead to fast adaptation. This is a computationally expensive procedure at both train and test time. Instead, in our work we learn amortized versions of the discriminator and policy models which removes this computational

overhead. Furthermore, the most challenging domain used in [12] is a modified Mountain-Car domain, a single-dimensional control problem.

Concurrent with [12], [34] extend IRL to the meta-learning setup using the MAML algorithm [4], a meta-learning method not too dissimilar to Reptile [22]. This work focuses on the setting where the observation space is high-dimensional, yet the state-action space is discrete and tractable. As a result of this choice, they use a model to predict the reward for every state-action pair in the task. Extending this work to intractable continuous control environments requires significant modifications to the algorithm, likely leading to a similar method to [12] which inherits the challenges described above.

The most similar prior work to our method is that of [33]. In this work, first a powerful VAE [19] is trained on the full dataset of expert demonstrations; a bi-directional LSTM model encodes a single trajectory and infers a posterior distribution, while a Wave-Net [24] decoder reconstructs the states and actions of this trajectory. After this pre-training step, the VAE is frozen. When training on a task from the task distribution, an expert trajectory is passed through the VAE encoder to obtain a sample from the posterior distribution, and the IRL algorithm as well as the policy condition on this sample. While this work demonstrates impressive experimental results, training good generative models of full trajectories is a difficult and expensive procedure that requires powerful models (consider for example image-based domains). Indeed, learning good dynamics models is a difficult, active problem studied by the model-based RL research community. Our work requires no such generative model to be trained.

Interestingly, concurrent with our work, [25] present a very effective Meta-*RL* algorithm that incorporates many similar choices to SMILe; a permutation-invariant context encoder is used to encode samples from a task's replay buffer, and the meta-policy learns to use this information to adapt itself more effectively. As there is no discriminator involved, the gradients from the Q function are used to update the encoder.

# 6 Conclusion & Future Work

In this work we propose SMILe, a novel framework for Meta Inverse Reinforcement Learning. Interpreting the Meta Imitation Learning problem from a context-conditional graphical model perspective, we develop to our knowledge the first efficient algorithm for Meta-IRL that scales to the function approximator setting. We demonstrated the effectiveness of our method on a series of difficult Meta Imitation Learning problems in high-dimensional continuous control, and observe that SMILe consistently outperforms Meta-BC. Furthermore, we observed that SMILe performed comparably or outperformed Meta-DAgger, while being applicable in the state-only setting and not requiring online experts. In future work we hope our framework and practical implementation details are applied to Imitation Learning from other forms of contexts such as linguistic input or images of desired goal states.

## Footnotes

[1] e.g. KL divergence or Wasserstein Distance

[2]We discuss exact details of dynamics randomizations in appendix E.2

[3]The reasoning behind this choice is discussed in Appendix E.1

[4]As defined in the original Walker task [2]

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
