[Supplementary Material · SMILe_Neurips_Supplementary_Final.pdf]

# A  SMILe Algorithm

## A.1  Algorithm Boxes

The full Meta-IRL algorithm for SMILe is described in Algorithm 1, where Algorithms 2 and 3 denote discriminator and policy update subroutines respectively. We find it crucial to regularize the discriminator model using a gradient penalty term [14] which we weight by the default factor of 10. $\mathcal{L}_{GP}$ in the Algorithm box below refers to this regularization term, and $\mathcal{L}_{disc}$ denotes the discriminator objective. $\lambda_{GP}$ is set to 10 for every SMILe model trained in this work. In the UpdateDiscriminator function, sampling transitions has a different meaning depending on whether we operate in the state-only or State-Action setting; in the state-action case transitions are $(s, a)$ pairs and in the state-only case, transitions are state-next-state pairs $(s, s')$.

---

**Algorithm 1:** SMILe

---

Meta-Policy $\pi_\theta$
Meta-Discriminator $D_\psi$
Context Encoder of Disciminator $Enc_\phi$
Meta-Training tasks $\{\mathcal{T}_i\}_{i=1}^N$
Aquire expert demonstrations per task $\{\mathcal{B}_i^{\text{exp}}\}_{i=1}^N$
Initialize per task replay buffers $\{\mathcal{B}_i\}_{i=1}^N$
// Populate buffers with initial rollouts
**for** *every* $\mathcal{T} \in \{\mathcal{T}_i\}_{i=1}^N$ **do**
    **for** *j times* **do**
        $\mathcal{E} \leftarrow$ Environment for task $\mathcal{T}$
        Sample a context $\mathcal{C}$ for task $\mathcal{T}$
        Do rollout with policy $\pi_\theta(a|s, Enc_\phi(\mathcal{C}))$ inside environment $\mathcal{E}$
        Add rollout to replay buffer for task $\mathcal{T}$
    **end**
**end**
// Train loop
**while** *not converged* **do**
    **for** *m times* **do**
        $\mathcal{T} \sim \text{Unif}(\{\mathcal{T}_i\}_{i=1}^N)$
        $\mathcal{E} \leftarrow$ Environment for task $\mathcal{T}$
        Sample a context $\mathcal{C}$ for task $\mathcal{T}$
        Do rollout with policy $\pi_\theta(a|s, Enc_\phi(\mathcal{C}))$ inside environment $\mathcal{E}$
        Add rollout to replay buffer for task $\mathcal{T}$
    **end**
    **for** *d times* **do**
        UpdateDiscriminator()
        UpdatePolicy()
    **end**
**end**

---

## A.2  Context Encoder

Inspired by the use of permutation-invariant encoders in recent meta-learning literature [10, 13], our encoder is composed of two models, one encoding individual transitions $(s, a, s')$, and a second aggregating these embeddings. Let `TransitionEncoder` $: \mathcal{S} \times \mathcal{A} \times \mathcal{S} \to \mathbb{R}^d$ and `Aggregator` $: \mathbb{R}^d \to \mathbb{R}^k$ both be embedding functions modelled by MLPs with linear output layers[5]. Let $\mathcal{C}$ be a context composed of multiple expert trajectories $\{\tau^i\}_{i=1}^P$. Let $(s_t^i, a_t^i, s_{t+1}^i)$ be the state, action, and next state of timestep $t$ for trajectory $i$. We use

$$v_{\mathcal{C}} = \texttt{Aggregator}\Big( \sum_{i=1}^P \frac{1}{T} \sum_{j=1}^T \texttt{TransitionEncoder}(s_t^i, a_t^i, s_{t+1}^i) \Big) \tag{7}$$

**Algorithm 2:** UpdateDiscriminator

---

Uniformly sample a batch of tasks $\{\mathcal{T}_i\}_{i=1}^K$
$\{\text{Tr}_i^{\exp}\}_{i=1}^K \leftarrow$ Sample $M$ expert transitions per task from provided demonstrations
$\{\text{Tr}_i^\pi\}_{i=1}^K \leftarrow$ Sample $M$ policy transitions per task from replay buffers
**for** $\mathcal{T}_i \in \{\mathcal{T}_i\}_{i=1}^K$ **do**
  |   Sample context $\mathcal{C}_i$ for task $\mathcal{T}_i$
**end**
```
// Discriminator objective term
```
$\mathcal{L}_{disc}^{1:K} \leftarrow \frac{1}{K}\sum_i \mathcal{L}_{disc}\left(\{\text{Tr}_i^{\exp}\}_{i=1}^K, \{\text{Tr}_i^\pi\}_{i=1}^K, Enc_\phi(\mathcal{C}_i)\right)$
```
// Gradient penalty term
```
$\mathcal{L}_{GP}^{1:K} \leftarrow \frac{1}{K}\sum_i \mathcal{L}_{GP}\left(\{\text{Tr}_i^{\exp}\}_{i=1}^K, \{\text{Tr}_i^\pi\}_{i=1}^K, Enc_\phi(\mathcal{C}_i)\right)$
$\psi \leftarrow \psi - \alpha_\psi \nabla_\psi \left(\mathcal{L}_{disc}^{1:K} + \lambda_{GP}\mathcal{L}_{GP}^{1:K}\right)$
$\phi \leftarrow \phi - \alpha_\phi \nabla_\phi \mathcal{L}_{disc}^{1:K}$

---

**Algorithm 3:** UpdatePolicy

---

Uniformly sample a batch of tasks $\{\mathcal{T}_i\}_{i=1}^K$
$\{\mathcal{C}_i\}_{i=1}^K \leftarrow$ Sample context $\mathcal{C}_i$ for each task
$\{\text{Tr}_i^\pi\}_{i=1}^K \leftarrow$ Sample $M$ transitions $(s, a, s', terminal)$ per task from the task replay buffers
To each sampled transition add the corresponding reward computed as
  $r \leftarrow \log D_\psi(s,a) - \log\left(1 - D_\psi(s,a)\right)$ or $r \leftarrow \log D_\psi(s,s') - \log\left(1 - D_\psi(s,s')\right)$
  depending on whether we operate in the state-action or state-only setting
Using transitions in $\{\text{Tr}_i^\pi\}_{i=1}^K$ update the policy, Q function, and Value function parameters using
  Soft-Actor-Critic update step

---

| Method | 4 demos, 20 sub | 16 demos, 20 sub | 64 demos, 20 sub | 64 demos, 1 sub |
|---|---|---|---|---|
| **Meta-BC (MSE)** | $0.931 \pm 0.620$ | $0.643 \pm 0.410$ | $0.468 \pm 0.274$ | $0.340 \pm 0.188$ |
| **Meta-BC (MLE)** | $1.481 \pm 0.859$ | $1.467 \pm 0.832$ | $1.281 \pm 0.798$ | $1.488 \pm 0.864$ |
| **SMILe (State-Action)** | $\mathbf{0.181 \pm 0.095}$ | $0.125 \pm 0.060$ | $\mathbf{0.102 \pm 0.100}$ | $\mathbf{0.118 \pm 0.078}$ |
| **SMILe (State-Only)** | $0.186 \pm 0.107$ | $\mathbf{0.113 \pm 0.058}$ | $0.121 \pm 0.086$ | $0.138 \pm 0.074$ |

Table 3: In the HalfCheetah Random Velocity task, the MSE variant of the baseline performs significantly better than the MLE variant.

as the representation for $\mathcal{C}$. As this embedding method is fully differentiable, the encoder and discriminator can be trained end-to-end with backpropagation.

### A.3 Soft-Actor-Critic

As our RL algorithm we chose to use Soft-Actor-Critic (SAC) [16]. All optimization parameters used were default except for the reward scale which was lightly tuned per experiment. We used the policy parameterization where given a state, the policy outputs mean and diagonal covariance parameters for a `Tanh(Normal(`$\mu(s), \Sigma(s)$`))` distribution. We used the double Q function approach. The target value function was updates in the "soft manner" using the default weight `0.005`.

## B Meta-BC and Meta-DAgger

As mentioned, for each experiment, the Meta-BC baseline is obtained by composing the SMILe encoder (Section A.2) and policy models. Meta-BC model are trained end-to-end with either the maximum-likelihood[6] or mean-squared error objective, depending on which results in better performance; we found that for the HalfCheetah Random Velocity and Ant 2D Goal experiments the MSE objective resulted in better policy performance (as demonstrated in Table 3 and Figure 3).

Meta-BC (MLE)          Meta-BC (MSE)

Figure 3: In the Ant 2D Goal experiment, the Meta-BC baselines perform better when trained with the MSE objective than with the MLE objective.

## B.1 Meta-BC Algorithm

The Meta-BC training procedure is described in Algorithm 4. $\mathcal{L}_{obj}$ may refer to either the maximum-likelihhod or mean-squared error objectives.

---

**Algorithm 4:** Meta-BC

---

Meta-Policy $\pi_\theta$
Context Encoder $Enc_\phi$
Meta-Training tasks $\{\mathcal{T}_i\}_{i=1}^N$
Aquire expert demonstrations per task $\{\mathcal{B}_i^{\text{exp}}\}_{i=1}^N$
// Train loop
**while** *not converged* **do**

> Uniformly sample a batch of tasks $\{\mathcal{T}_i\}_{i=1}^K$
> $\{\mathcal{C}_i\}_{i=1}^K \leftarrow$ Sample context $\mathcal{C}_i$ for each task
> $\{\text{Tr}_i\}_{i=1}^K \leftarrow$ Sample $M$ state-action pairs per task from the expert denomstrations
> $\mathcal{L}_{obj}^{1:K} \leftarrow \frac{1}{K}\sum_i \mathcal{L}_{obj}\left(\{\text{Tr}_i\}_{i=1}^K, Enc_\phi(\mathcal{C}_i)\right)$
> $\psi \leftarrow \psi - \alpha_\psi \nabla_\psi \left(\mathcal{L}_{obj}^{1:K}\right)$
> $\phi \leftarrow \phi - \alpha_\phi \nabla_\phi \mathcal{L}_{obj}^{1:K}$

**end**

---

## B.2 Meta-DAgger

The Meta-DAgger algorithm is very similar to the Meta-BC algorithm. The main difference is that after each epoch we generate a set of rollouts using the policy at that point. The states visited in these rollouts are labelled with an action using the expert policies and added to the dataset for performing supervised learning (Behavioural Cloning).

# C HalfCheetah Random Velocity

## C.1 Data

The expert for this task was trained using SAC [16] by concatenating the target velocity to the observations. As noted we evaluated different models by training on various amounts of expert demonstrations. At each data amount, we normalized the states of meta-train demos, meta-test demos, and the environment using the mean and standard deviations computed from the meta-train demos.

## C.2 Evaluation Protocol

Each model is trained with 4 random seeds on each dataset size. To evaluate each run, per eval task, for every context size from 1 to 4, for 3 different samples of the context, we do 5 rollouts with th conditional policy. For each timestep of every rollout, the absolute difference between the velocity and target velocity is computed. All such computed values are averaged to obtain the evaluation metric per random seed. Table results show the mean and std of this values across seeds.

## C.3 Dependence on Context Size

Figure 4 demonstrates that for this task, additional gains are not made from conditioning on more expert demonstrations.

Figure 4: Delta from target velocity given different context sizes. For HalfCheetah random velocity problem, there is little dependence on the number of context trajectories provided to the model. This plot contains results when models are train using the largest amount of expert demonstrations (64 demonstrations per task with subsampling 1 (i.e. no subsampling)). Results are similarly flat regardless of the amount of expert demonstrations used to train models. Due to the mistake detailed in the caption of Table 3, the Meta-BC results do not match those in the table presented in the main document. As noted in the caption of Table 3, this mistake will be fixed in the main document given first opportunity.

# D Ant 2D Navigation

## D.1 Data

Separate experts for each task were trained using SAC [16]. As noted we evaluated different models by training on various amounts of expert demonstrations. At each data amount, we normalized the states of meta-train demos, meta-test demos, and the environment using the mean and standard deviations computed from the meta-train demos.

## D.2 Ant 2D Navigation Evaluation Protocol

Each model is trained with 4 random seeds per dataset size. We evaluate each seed as follows: For each meta-test direction, sample 3 contexts. For each context do 10 rollouts with the conditional policy and compute the closest the agent comes to the target. The mean of these 30 rollouts is the performance of the seed for that target. Plots show mean and std of these values.

## D.3 RL is Not Sufficient

To demonstrate the Imitation Learning is necessary for solving this task, we train an RL algorithms using SAC [16] that receives as part of its state the target goal location. The reward function is 1.0

Figure 5: An RL algorithm that observes as part of its state the goal location cannot solve this task due to the sparsity of the rewards. Imitation Learning is necessary to overcome the exploration problem.

if the agent is within 0.5 radius of the target and 0 otherwise. Figure 5 demonstrates that indeed Imitation Learning is necessary to overcome the exploration problem in this task.

## E  Walker Random Dynamics

### E.1  Data

To generate expert demonstrations, we initially experimented with training a single expert policy that receives as part of its state representation the randomization parameters used. However, we observed that the final performance of this expert varied significantly on different environments. This could be due to the difficulty of the particular dynamics, or due to subpotimal training of the expert policy. To remove this confounding factor, we decided to train separate expert policies for each of the meta-train and meta-test tasks. We normalized the states of meta-train demos, meta-test demos, and the environment using the mean and standard deviations computed from the meta-train demos.

### E.2  Task Design

The dynamics randomizations we perform are identical to the ones found in, `https://github.com/dennisl88/rand_param_envs`, also used in [25]. Note that the Walker environment in this repository uses a timestep rate of 5 instead of the typical 4. We also use 5 timestep rate and keep this in mind when training all models, including experts and the baseline policy used for our evaluation metric.

### E.3  Evaluation Protocol

Both variants of SMILe and the Meta-BC baseline were trained using 3 random seeds. Each random seed of each approach was evaluated on the meta-test dynamics. Per dynamics settings, 4 contexts were samples and per context the conditional policy was used to generate 4 trajectories. The average return obtained was used as the performance of that random seed on that task/dynamics.

## F  Ant Linear Classification

### F.1  Data

As discussed, expert demonstrations were generated using two of the experts trained for the Ant 2D Goal experiment. We normalized the states of meta-train demos, meta-test demos, and the environment using the mean and standard deviations computed from the meta-train demos.

### F.2  Evaluation Protocol

Both variants of SMILe and the Meta-BC baseline were trained using 3 random seeds. Each random seed of each approach was evaluated on the meta-test tasks. For each meta-test task, for 4 times we did the following: We sampled 12 expert trajectories. For every $i$ from 1 to 12 we did the following: we took the first $i$ trajectories as the context, and evaluated the conditional policy on 20 rollouts. The success rate from all these rollouts was considered the success rate on this meta-test task.

| Architectures & Hyperparameters | HalfCheetah Rand Vel | Ant 2D Goal | Walker Random Dynamics | Ant Linear Classification |
|---|---|---|---|---|
| `TransitionEncoder` out dim | 64 | 64 | 64 | 128 |
| `TransitionEncoder` arch | 2 lay - 256 dim - relu - bn | 2 lay - 256 dim - relu - bn | 2 lay - 256 dim - relu - bn | 2 lay - 256 dim - relu - bn |
| `Aggregator` out dim | 64 | 64 | 64 | 64 |
| `Aggregator` arch | 1 lay - 64 dim - relu - bn | 1 lay - 64 dim - relu - bn | 2 lay - 64 dim - relu - bn | 2 lay - 64 dim - relu - bn |
| Disc arch | 3 lay - 512 dim - relu | 3 lay - 512 dim - relu | 3 lay - 512 dim - relu | 3 lay - 1024 dim - relu |
| Policy arch (Q, V fn's same) | 3 lay - 256 dim - relu | 3 lay - 512 dim - relu | 3 lay - 512 dim - relu | 3 lay - 256 dim - relu |
| Expert Policy arch (Q, V fn's same) | 3 lay - 256 dim - relu | 3 lay - 256 dim - relu | 2 lay - 256 dim - relu | 3 lay - 256 dim - relu |
| Rollouts per task before training | 1 | 125 | 25 | 16 |
| Rollouts between update cycles | 5 | 32 | 25 | 8 |
| Max episode length | 1000 | 100 | 1000 | 50 |
| Replay buffer size per task | 50000 | 64000 | 50000 | 2500 |
| Number of updates per cycle | 100 | 400 | 1000 | 50 |
| Number of tasks used per update | 8 | 8 | 10 | 8 |
| Discriminator batch size per task | [80,128] | 250 | 100 | [144,256] |
| Policy batch size per task | 128 | 128 | 100 | 256 |
| SAC reward scale | 24.0 | 4.0 | 20.0 | 4.0 |
| Discriminator Logit Clamp Range | [-10,10] | [-10,10] | [-10,10] | [-10,10] |

Table 4: Various hyperparameters and model sizes used for experimental results

# G  Hyperparameters

## G.1  Optimization Parameters

For all SMILe models we use the Adam optimizer [18] with a learning rate of 3e-4, and default $\beta_2 = 0.999$ parameter. However, we find it beneficial to use different $\beta_1$ parameters for the different models; for the discriminator we set $\beta_1$ to 0, for the encoder to 0.9, and for the policy to 0.25. These $\beta_1$ parameters were chosen through a small hyperparameter searches at early stages of this work, tuned on toy proof-of-concept tasks. Works such as [11] may provide additional insight. The gradient penalty regularization term was weighted by a factor of 10.

For the Meta-BC and Meta-DAgger we use the Adam optimizer [18] with learning rate 3e-4, $\beta_1 = 0.9$, and $\beta_2 = 0.999$. We also tried $\beta_1 = 0$ and $\beta_1 = 0.25$ but did not obtain gains.

## G.2  Table of Hyperparameters

As there are many model architectures and hyperparameters to consider, we performed very little (or none) tuning of most of the choices outlined in Table 4. When implementing a new experiment, we would mainly tune the SAC reward scale, discriminator architecture, policy architecture, and encoder architecture, typically in that order. For the Ant 2D Goal task, we found that number of "Rollouts per task before training" should not be small (e.g. 1-5), but we don't expect such a large number of 125 to be necessary either. We also made very little attempt at searching over dimensionality of model layers. It is almost certain that much smaller models could perform as well as ours.

The MLPs used for `TransitionEncoder` and `Aggregator` end with an additional linear fully connected layer mapping to the output dimension. The MLPs used for policy architectures end with two separate linear fully connected layers for outputting the mean and diagonal covariance of the action distributions.

For the HalfCheetah Random Velocity and Ant Linear Classification tasks, the "Discriminator batch size per task" entry in the table is a range since the context size for these experiments is a variable number of expert trajectories.

## Footnotes

[5]We found linear output layers to be important for `TransitionEncoder` in particular

[6]Note that given a state $s$, the policy output is the distribution `Tanh(Normal(`$\mu(s), \Sigma(s)$`))`