[Reviews · NeurIPS 2019]

Reviewer 1



The paper introduces a scalable approach for doing meta inverse RL based on maximum entropy IRL. The baseline is a meta-learning method based on behavioral cloning over which a significant performance improvement is obtained, Pro: The approach seems technically sound, building on the theory of AIRL/GAIL. Also, implementing the equations in a practical and efficient way is a non-trivial contribution. Furthermore, the paper is clearly written. The motivation for IRL versus BC and the advantages that IL can have over RL are clearly explained. Con: The evaluation of the method could be more extensive. Specifically, a more detailed comparison with meta-RL methods and existing meta-IRL methods would help in better evaluating the strengths of the method. Currently, only an empirical comparison is made against a meta-BC baseline. Comparing against meta-RL methods is relevant, because the paper motivates the use of IL by the fact that it can be used on complex tasks where RL methods fail. While this is true in principle, the domains used in the experiment are all domains where standard RL methods can obtain a good performance. Comparing against meta-IRL methods is also relevant, in particular, with the method from citation [33]. The main advantage of SMILe over this method appears to be a computational one. However, it remains unclear how much this computational advantage is for the domains considered in this paper. Furthermore, it would be interesting to also compare the sample efficiency of the two methods. This gives insight into the cost (in terms of sample efficiency) that SMILe pays for the gain (in terms of computation). Overall: Interesting, new meta-IRL method with limited empirical evaluation. Minor: line 39/44: what does "intractable" mean in this context. If it just refers to general function approximation, I would drop this term. AFTER AUTHOR REBUTTAL: The additional experiments from the author response have (partly) addressed my concerns. I've increased my score by 1, on the condition that these experiments are added to the final paper.

Reviewer 2



The submission is well-written with sufficient experiments supporting the argument. The proposed method is clear and convincing. The reviewer is just concerned with the significance of the submission. It seems that the biggest difference of the proposed method compared with AIRL frame work lies on the off-policy training, which however is borrowed from [20] and also SEEMINGLY related to Rakelly, Kate, et al (2019). Therefore, it would be great if the author can further emphasize the differences of the proposed method compared with all these methods in the rebuttal. Rakelly, Kate, et al. "Efficient off-policy meta-reinforcement learning via probabilistic context variables." arXiv preprint arXiv:1903.08254 (2019). ================================================================== The rebuttal of authors is clear and convincing. Therefore, I increase my score to 7.

Reviewer 3



The quality of the submission is fairly high; the writing is good and the paper is easy to follow and feels complete. The originality of the work is just OK; it is a fairly straightforward idea and can be boiled down to AIRL with conditioning''. Significance is reasonably high given the current popularity of meta-learning and IRL. No theory is provided but the experiments have good coverage, testing a number of different aspects of the proposed method, and the methodology seems sound. One exception to this is the rather weak behavior cloning baseline. One way in which the exposition could be improved is by being more explicit about how different functions are parameterized, and being more explicit about how the discriminator is optimized. At this point in the history of AI I guess it's safe to assume you performed gradient ascent on the objective in equation 4, but it's probably still worth being explicit about that. line 87: shouldn't there by an entropy term in the reward function? In Figure 1, it feels like there should be an arrow from E to C, especially since for all tasks considered in this paper, C takes the form of expert trajectories, which are of course heavily dependent on E (e.g. beginning of section 3.3). E and C are definitely not independent of one another given T. Line 250: did you mean \exp_base instead of exp_base? or maybe \pi_base? exp_base seems out of line with the rest of the notation. The subsampling of trajectories, discussed briefly in the caption of Table 1, is not really addressed in the main text, but probably should be as its role was not clear to me. Would be interested in seeing results for non-trivial generalization (e.g. extrapolative generalization rather than interpolative), and in more difficult stochastic domains. The description of the classification experiment in Section 4.4 should be made clearer. I'm confused about what the classification decision is supposed to be based on, given that the agent is provided 2 points, one on either side of the hyperplane for the task. Is it based on the order of the two points? (i.e. if the first point is on side A of the hyperplane, go to location 1, if it's on side B of the hyperplane go to location 2). Also I'm confused about how you are able to use the experts from Ant 2D goal as the experts in this task, since the state space is now augmented with the 2 4-dimensional vectors to be classified...I guess you just choose which expert to use based which target the expert is supposed to go to, and have the chosen expert ignore the 4-dimensional vectors? To provide context for the level of performance achieved in this task, it might be useful to see performance of a linear binary classifier trained under the same conditions (e.g. seeing only 1 to 8 pairs of training points per task).

[Author Response · NeurIPS 2019]

| Method | HC Rand Vel | Walker Returns | Walker Custom Metric | Ant Lin Class Success Rate |
|---|---|---|---|---|
| **Meta-DAgger** | **0.082 ± 0.059** | 2404.1 ± 91.7 | 0.275 ± 0.089 | **0.67 ± 0.00** |
| **SMILe (state-action)** | **0.118 ± 0.078** | **3106.6 ± 67.6** | **0.710 ± 0.036** | **0.68 ± 0.01** |

Table 1: Meta-Dagger results compared to SMILe (SMILe results taken from the paper)

Figure 1: Ant Random Goal, Final Distance to Goal: Meta-DAgger (Left), SMILe (Middle), Fully Observed RL (Right)

We thank the reviewers for their careful reading and constructive comments, and have already updated the manuscript.

**New Experiments**: **R1, R4 (Stronger Meta-Imitation Baseline)**: As an additional baseline for Meta Imitation
Learning (Meta-IL) we implemented a meta variant of DAgger and tuned it carefully. The implementation of Meta-
DAgger is similar to Meta Behavior Cloning with the difference that in each epoch expert policies are used to label an
additional set of policy rollouts. Importantly, Meta-DAgger operates under the much stronger assumption than SMILe
that expert policies are always available to give the correct actions for states visited by the student policy. Meta-Dagger
therefore serves as one of the strongest Meta-IL baselines. As shown in Table 1 and Figure 1, SMILe's performance is
comparable to Meta-DAgger in all tasks, except for Walker Random Dynamics where SMILe achieves significantly
better performance.

**R1, R4 (Meta-RL Baseline)**: One of the fundamental advantages of Imitation Learning compared to standard RL is in
sparse reward tasks. As such, in the Ant 2D Goal task we trained "fully observed" policies (using SAC) which observe
task parameters as part of the state; such policies are strictly stronger than any Meta-RL baseline such as MAML or
PEARL. The reward function is $1.0$ if the agent is within $0.5$ radius of the target and $0$ otherwise. As shown in the
Figure above, fully observable policies were not able to solve this sparse reward task. We will incorporate these as well
as Meta-DAgger results in our manuscript.

**Addressing Other Comments R1,R3,R4 (Significance of Contribution)**: To our knowledge, the only prior Meta-IRL
method that scales to the function approximator setting is the work of Wang et al. The relative contributions of our
method are not limited to computational gains. Importantly, **SMILe does not require learning a generative model**
**over trajectories**. Wang et al. train a VAE on expert trajectories, where a decoder must learn to both imitate expert
policies and learn a good generative model of environment dynamics. Learning good dynamics models is a difficult,
active problem studied by the model-based RL research community. This problem will become even harder as the
community seeks to scale meta-IRL methods to domains with richer observation spaces (e.g. image observations).

**R3 (Off-policy as Contribution)**: We do not consider off-policy training a core contribution of our work. It is merely
used to make training of our models significantly faster in terms of wall-clock time. Lastly, we agree that our method
does have a number of similarities to the Meta-*RL* work of Rakelly et al as discussed in Related Work. We note that
their work is concurrent (first posted on arxiv 2 months prior to Neurips deadline), and they are addressing the Meta-*RL*,
not the Meta-IRL (Meta Imitation Learning) problem.

**R4 (Testing Harder Generalization)**: In the Walker Random Dynamics task, during training models observed 50
random settings of dynamics variables. Generalizing to the 25 testing dynamics is highly non-trivial. SMILe's
respectable performance gives us confidence about the quality of adaptive policies learned, given our new result that
Meta-DAgger performs substantially worse than SMILe on this task. We believe these results demonstrate that SMILe
is an important contribution to Meta Imitation Learning. **R4 (Clarifying Ant Linear Classification)**: R4 is correct
in their interpretation of our task setup. As part of their state, agents observe two 4-dimensional vectors. If the first
point is from the positive class the agent should navigate to the first target, otherwise it should navigate to the second
target. When generating expert demonstrations we use our knowledge of the correct label to choose an appropriate
expert policy from the Ant 2D Goal task, which ignores the two 4-dimensional points in the state. We have updated
Section 4.4 in our manuscript with these clarifications. **R4 (Connecting E to C in Graphical Model)**: The graphical
models in our manuscript describe the generating process for states and actions under a policy $\pi$. If we know the
task identity (T observed), knowing the environment (E observed) the policy is being rolled out in does not provide
additional information about what context (C) the policy conditioned on. In contrast, if T was not observed, E would
tell us that the policy conditioned some context coming from a task that is compatible with E. Hence we do not believe
there should be an arrow from E to C. **R4 (Parameterizations)**: All models (encoder, discriminator, policy, Q, and
V) architectures are provided in the Appendix G.2 of our manuscript. We used the Adam optimizer throughout. **R4**
**(Entropy in Policy Objective)**: In contrast to GAIL, the AIRL algorithm does not include a causal entropy term. **R4**
**(exp_base Notation)**: We used exp_base to denote the expert policy for the Walker environment with unmodified
dynamics. We will remove this notation from subsequent revisions as there is no need to name this policy explicitly.

[Meta-Review · NeurIPS 2019]

The originality and quality are nice. The main contribution is the introduction of a scalable, meta, inverse RL method, which is quite important and hot topic in IRL. Moreover, the additional experiments from the author response would be good enough to accept. So, I recommend the paper be accepted.